# Personalized Nutrition in the Management of Female Infertility: New Insights on Chronic Low-Grade Inflammation

**DOI:** 10.3390/nu14091918

**Published:** 2022-05-03

**Authors:** Gemma Fabozzi, Giulia Verdone, Mariachiara Allori, Danilo Cimadomo, Carla Tatone, Liborio Stuppia, Marica Franzago, Nicolò Ubaldi, Alberto Vaiarelli, Filippo Maria Ubaldi, Laura Rienzi, Gianluca Gennarelli

**Affiliations:** 1B-Woman, 00197 Rome, Italy; verdone@b-woman.it (G.V.); allori@b-woman.it (M.A.); 2Clinica Valle Giulia, GeneraLife IVF, 00197 Rome, Italy; cimadomo@generalifeitalia.it (D.C.); vaiarelli@generalifeitalia.it (A.V.); ubaldi@generalifeitalia.it (F.M.U.); rienzi@generalifeitalia.it (L.R.); 3Department of Life, Health and Environmental Sciences, University of L’Aquila, 67100 L’Aquila, Italy; carla.tatone@univaq.it; 4Center for Advanced Studies and Technology (CAST), University “G. d’Annunzio” of Chieti-Pescara, 66100 Chieti, Italy; stuppia@unich.it (L.S.); marica.franzago@unich.it (M.F.); 5Department of Psychological, Health and Territorial Sciences, School of Medicine and Health Sciences, University “G. d’Annunzio” of Chieti-Pescara, 66100 Chieti, Italy; 6Department of Medicine and Aging Sciences, University “G. d’Annunzio” of Chieti-Pescara, 66100 Chieti, Italy; 7Radiology Unit, Department of Medical Surgical Sciences and Translational Medicine, Sant’Andrea Hospital, Sapienza University of Rome, 00189 Rome, Italy; ubaldi.nicolo@gmail.com; 8Department of Biomolecular Sciences, University of Urbino “Carlo Bo”, 61029 Urbino, Italy; 9Livet, GeneraLife IVF, 10126 Turin, Italy; gennarelligl@gmail.com

**Keywords:** precision nutrition, infertility, nutrigenetic, nutrigenomics, epigenetics, microbiota, chronic low-grade inflammation

## Abstract

Increasing evidence on the significance of nutrition in reproduction is emerging from both animal and human studies, suggesting a mutual association between nutrition and female fertility. Different “fertile” dietary patterns have been studied; however, in humans, conflicting results or weak correlations are often reported, probably because of the individual variations in genome, proteome, metabolome, and microbiome and the extent of exposure to different environmental conditions. In this scenario, “precision nutrition”, namely personalized dietary patterns based on deep phenotyping and on metabolomics, microbiome, and nutrigenetics of each case, might be more efficient for infertile patients than applying a generic nutritional approach. In this review, we report on new insights into the nutritional management of infertile patients, discussing the main nutrigenetic, nutrigenomic, and microbiomic aspects that should be investigated to achieve effective personalized nutritional interventions. Specifically, we will focus on the management of low-grade chronic inflammation, which is associated with several infertility-related diseases.

## 1. Introduction

### 1.1. Association between Nutrition and Fertility

Infertility is a disease defined by the failure to achieve pregnancy after 12 or more months of regular unprotected sexual intercourse [1]. Infertility affects 48.5 million couples worldwide, with important psychological implications for the couple and with a negative impact on the quality of their life [2]. Assisted reproductive technologies (ART) represent the most effective mean to treat infertility. However, despite significant and constant advances in ART, success rates only marginally increased across the decades [3,4]. Since lifestyle and environmental factors such as alcohol and caffeine consumption, smoking, nutritional habits, pesticides, and endocrine disruptors seem to exert a profound impact on reproductive health [5], lately several efforts were made to investigate whether modifiable habits in parental lifestyle, particularly maternal nutrition, can be targeted for a better reproductive outcome.

Nutrition has been associated with the development of multiple conditions [6], and mounting evidence suggests an interdependent correlation between nutrition and female fertility [7]. Improper food consumption, leading to unbalanced caloric intake, is responsible for abnormal body weight. Several studies have shown how body mass index (BMI) has a J-shaped correlation with the risk of infertility: both underweight (BMI < 19 kg/m^2^) and overweight (BMI 25–29.9 kg/m^2^) women have a similar risk of infertility [8,9,10]. This is because either poor or excessive intake of micro and macro nutrients such as carbohydrates, proteins, vitamins, and minerals alter energy balance, which is directly correlated to reproductive performance [8].

The hypothesis that an appropriate diet improves fertility is supported by studies in both animal models and humans. In particular, the Mediterranean diet (MedDiet) has been studied in this regard [11,12,13,14,15], but many studies also investigated the effect of dietary intake of specific macronutrient(s) or micronutrient(s) (such as proteins, fats, carbohydrates, vitamins and minerals such as vitamin B12, vitamin D, folates zinc, omega-3) with the risk of infertility [16,17,18]. Although conflicting results exist for diary consumption [19,20], female reproductive health, in general, seems to benefit from a correct balance of proteins, carbohydrates, lipids, antioxidants, and folate in the daily diet. In particular, the consumption of whole grains, fruits, vegetables, fish (rich in omega-3 polyunsaturated fatty acids (PUFAs)), olive oil (rich in monounsaturated fatty acids (MUFAs)), and low consumption of trans-fats may not only improve overall health but also enhance fertility [17]. Moreover, an adequate intake of antioxidants, folic acid, β-carotene, vitamin C, E, and especially folates and choline for supporting the one-carbon metabolism (1-C), namely a series of interlinking metabolic pathways comprising folate cycle, methionine remethylation, and trans-sulfuration [21,22], are associated with shorter time to pregnancy [7].

### 1.2. From General Population-Based Recommendations to Precision Nutrition for Infertility

Whereas the literature on the correlation between diet and fertility is steadily increasing, and the evidence that a strong link between unhealthy dietary habits and infertility is undeniable, there are no official guidelines on the nutritional management of patients seeking a pregnancy. Likewise, IVF is not routinely combined with a nutritional counseling program.

Indeed, no specific dietary patterns for improving reproductive chances have yet been identified. This is mainly due to: (i) conflicting results reported from different studies; (ii) limited sample size; (iii) heterogeneity of populations under study; (iv) different confounding factors interfering with the correlation diet-reproductive outcomes; (v) self-reporting in the methods adopted for dietary assessment, such as food frequency questionnaires (FFQs). Although FFQs represent the most practical way to assess dietary habits, in large prospective studies, they are insufficient to accurately evaluate diet composition. Since follicle depletion is an ongoing process that begins during fetal development and continues throughout a woman’s reproductive life, assessing the correlation between diet and reproduction would be easier if the whole nutritional and lifestyle “story” of a woman is examined instead of just focusing on the situation at the time of conception, by means of FFQ with all its inherent limitations [22].

Finally, the most important aspect: all these studies used a “one size fits all” principle in analyzing stratified nutrition intervention, the most used method for giving general guidelines to groups of individuals who share key characteristics (e.g., “eat at least five portions of fruit and vegetables daily”) [23]. Of note, such an approach is mostly intended to limit/avoid deficiencies rather than to improve health [24]. Everyone has its own genome, proteome, metabolome, microbiome, and exposome, namely the totality of exposure over the lifetime, which has been demonstrated to affect the genetics, epigenetics, and immune system in humans [25]. Therefore, the environment must be considered since it influences, together with genetic variability, the way dietary components are absorbed, metabolized, and utilized [26]. For this reason, it is utopian, if not impossible, to determine a dietary pattern that fits all patients in relation to the treatment of infertility or any other disease.

As for all branches of medicine, a personalized and “precise” approach should be applied for the nutritional support of infertile patients [27]. Diet should be tailored to the individual just like the pharmacological and therapeutical treatments for infertility are adapted to each woman’s characteristics (deep phenotyping, nutrigenetics, microbiome, etc.) [28,29]. This approach is known as “precision nutrition” [24], and it might be more effective than general dietary advice [30,31].

### 1.3. The Emerging Role of Chronic Low-Grade Inflammation in Infertility

The mechanisms linking parental periconceptional diet to reproductive health are not fully understood. Eventually, several hormonal imbalances are involved, leading to the dysregulation of both the HPG axis and gonadal steroidogenesis [8]. A further crucial mechanism by which nutrition may affect reproductive function is the modulatory effect on inflammatory processes exerted by many nutrients and non-nutrient food components [32].

Inflammation is an innate defense response of the microcirculation occurring after injury or infection in locally blood-supplied tissues, which activates immune cells and releases various soluble mediators such as chemokines, cytokines, eicosanoids (e.g., prostaglandins), free radicals, and vasoactive amines by resident cells (tissue macrophages, dendritic cells, lymphocytes, endothelial cells, fibroblasts, etc.) [33]. A correct inflammatory response consists of three main steps: (i) acute inflammatory response, namely the production of inflammatory mediators (e.g., cytokines) by resident cells, infiltration of leukocytes, elimination of pathogen and/or debris; (ii) resolution, namely the removal of inflammatory stimuli, catabolism of proinflammatory mediators, polymorphonuclear cells death and efferocytosis and influx of monocyte-derived macrophages; (iii) post-resolution, namely the influx of adaptive immune cells, re-assembly of tissue-resident macrophages and dendritic cells, the establishment of adaptive immunity. The occurrence of all these phases is required to restore the functional homeostasis and the transition from innate (rapid, non-specific inflammatory response) to adaptive (more rapid and effective response to reinfection) immunity. However, when an incomplete resolution of the initial acute response occurs, a chronic inflammatory status is established. This, in turn, leads to the persistence of inflammatory triggers, which results in the persistence of inflammatory macrophages, chemokine, and cytokine synthesis/secretion and in the failure to establish adaptive immunity with a persistent level of tumor necrosis factor (TNF), interferons, and, most importantly, IL-6 [33].

Inflammation seems crucial in reproduction. Several reproductive processes, such as ovulation, menstruation, implantation, placentation, and pregnancy, depend on inflammatory pathways [34,35,36,37]. Therefore, dysregulation of either the magnitude or the duration of inflammatory events is strictly involved in the pathophysiology of infertility, and increasing evidence suggests that different diseases linked to infertility are related to chronic low-grade inflammation (Figure 1). For instance, patients affected by polycystic ovary syndrome (PCOS) show increased inflammatory markers, together with increased levels of *C-*reactive protein (CRP), IL-18, TNF-α, IL-6, white blood cell count (WBC), monocyte chemoattractant protein-1 (MCP-1), and macrophage inflammatory protein-1α (MIP-1α) [38]. Altered gene expression and genetic variants of TNF-α and IL-6 genes have also been suggested in PCOS [39,40]. In patients affected by endometriosis, also, elevated levels of inflammatory cytokines, such as IL-17, IL-6, and TNF-α, are found in the peritoneal fluid [41,42], and the nuclear factor-kB (NF-kB) seems responsible for the activation of the inflammatory process leading to the overexpression of p450 aromatase in the endometrium. This, in turn, increases the local production of estrogens, altering endometrial receptivity [41,42]. NF-kB also impacts ovarian physiology, altering the local intrafollicular environment and resulting in a larger production of intracellular reactive oxygen species (ROS) that impair oocyte competence [43]. Proinflammatory cytokines (IL-6, IL1β, interferon (IFN) α, TNF-α, IFNγ), as well as anti-inflammatory or regulatory mediators (IL-10, TGFβ), are increased even in the endometrium of women affected by adenomyosis, confirming the immunological changes associated with this disease [44]. Similarly, chronic endometritis (CE) is characterized by a low-grade local chronic inflammation with increased local concentrations of IL-1b and TNF-α [45]. At last, hydrosalpinx involves a significant increase in inflammatory cells in the endometrium, increased IL-2 concentrations, upregulation of NF-kB, and decreased expression of leukemia inhibitory factor (LIF), a member of the IL-6 cytokine family [34].

Patients with a diagnosis of unexplained infertility (UI) or premature ovarian failure (POF) also show an imbalanced adaptive immunity, with a persistent status of chronic inflammation [46]. Women with UI often display increased T helper 1/T helper 2 ratios and T helper 17 levels, both involving a proinflammatory state [47,48,49]. POF, as well, is characterized by a disequilibrium between anti-inflammatory and proinflammatory cytokines [50,51,52], which suggests that inflammation, aging, and premature ovarian insufficiency are closely related [52].

Chronic systemic inflammation is typical even across autoimmune diseases (ADs), often associated with complications of fertility [53,54,55]. A chronic inflammatory state often imbalances the immune microenvironment and results in the production of autoantibodies, possibly triggering ADs [53,55,56,57]. Among the most common: anti-phospholipid syndrome (APs) and systemic lupus erythematosus (SLE) [58], rheumatoid arthritis (RA) [59], autoimmune thyroid disease (AITD) [60], and celiac disease [55].

The pathway by which chronic low-grade inflammation impairs reproduction still needs to be fully elucidated. However, chronic inflammation may impair folliculogenesis via oxidative stress [61]. Indeed, inflammation and oxidative stress are mutually nourishing each other. In detail, inflammation promotes oxidative stress through increased NF-kB -p65 phosphorylation, the consequent increased expression of the redox family of NADPH oxidases (NOX), and the production of superoxide (O_2_), which is subsequently converted to hydrogen peroxide (H_2_O_2_) from the superoxide dismutase (SOD). ROS species (O^2−^ and H_2_O_2_) then freely move from the organelle to the cytoplasm and activate NF-kB -p65 phosphorylation, thereby increasing the expression of proinflammatory cytokines, including TNFα and IL-6 [62], and spreading inflammation.

Chronic low-grade inflammation may impair endometrial receptivity as well. In fact, inflammatory conditions such as endometriosis, adenomyosis, and CE figure among the main endometrial causes of recurrent pregnancy loss (RPL) and impair the chances of a full-term pregnancy via well-known mechanisms [63,64].

A further mechanism by which inflammation affects reproduction is by altering blood coagulation. Increased blood coagulation and thrombosis can result from activated immune conditions, including elevated proinflammatory cytokines (TNF-α, IL-1β, IL-6, IL-8), aberrant allo-immunity, and autoantibodies [65,66,67]. These effects occur even in the absence of genetic variants associated with maternal thrombophilia (e.g., factor XIII and factor II of coagulation and polymorphism in plasminogen activator inhibitor (PAI-1)). Blood coagulation is pivotal in embryo-endometrium interaction [68], and coagulation defects impair implantation [69,70,71,72]. As a matter of fact, altered coagulation is often reported in proinflammatory conditions such as endometriosis [73], PCOS [74], adenomyosis [75], and ADs [71,76,77,78], further confirming the interdependent correlation between chronic low-grade inflammation and blood coagulation.

Increasing evidence suggests that many nutrients and non-nutrient foods and components such as phytochemicals [79] modulate inflammation both acutely and in the long term [32,80]. It is therefore expected that a targeted “anti-inflammatory” nutritional support for infertile patients may represent a valuable tool to lower the proinflammatory status often associated with infertility. The putative negative effects exerted by both oxidative stress and altered blood coagulation on gamete, embryo, and endometrial competence might be thereby improved.

## 2. Tailoring the Nutritional Management of the Infertile Patient Using an “Anti-Inflammatory Approach”

### 2.1. Nutrigenetic Features Potentially Useful in the Management of Low-Grade Inflammation among Infertile Patients

The Human Genome Project made it possible to identify genetic variants involved in nutrient metabolism. This led to the development of nutrigenetics, the branch of science that investigates how the genotype influences the body’s response to food, nutrients, and nutrition-related diseases [81]. In the last decade, nutrigenetics rapidly developed and led to rapid growth in the number of companies offering direct-to-consumer genetically based testing (DTC-GT). These tests are not only aimed at estimating the risk of developing clinical conditions such as diabetes, cancer, or cardiovascular disease, but they also suggest “DNA diets”, namely diets personalized according to an individual’s genotype [82]. “Eat right for your genotype” is the tagline of most of these marketing strategies [83]. However, evidence-based advices or guidelines in this field are missing and a diet entirely based on the genotype lacks scientific support [84]. Genetic tests should be preferentially included within a functional strategy, correlating the results obtained with the patient’s clinical characteristics, symptoms, and diet habits [85]. Keeping these limitations in mind, these tests might be a valuable tool for better understanding everyone’s response to specific dietary or nutrient patterns. The professionals could benefit from these tests to develop more precise and effective dietary plans [24,86]. Single nucleotide polymorphisms (SNPs), which lead to a single base change in the DNA sequence, represent the simplest and most frequent kind of genetic variation used to implement nutrigenetic tests [83].

Hereafter, we will describe a list of SNPs that might be investigated for the nutritional management of infertile patients, aiming to outline possible nutritional strategies to manage low-grade inflammation.

#### 2.1.1. Folates and Choline Metabolism

Folic acid and folate are water-soluble B vitamins, also known as vitamin B9 [87]. The two terms are often used interchangeably, but the first refers to the synthetic molecule introduced in supplements or fortified foods, while the second is the form naturally present in some foods, including green and leafy vegetables, sprouts, some fruits, legumes, seeds, and offal [88].

Folates are essential for the synthesis of DNA and proteins, especially in tissues subject to processes of proliferation and differentiation [87]. Choline, sometimes referred to as vitamin J, is an amine that acts as a coenzyme in numerous metabolic reactions, and it is involved in the formation of cell membranes and in the synthesis of cholinic neurotransmitters, such as acetylcholine and the methyl group donor, betaine [89]. Natural dietary sources of choline are in both water-soluble and lipid-soluble forms, with high concentrations in milk, liver, egg yolk, meat, and wheat germ [89].

Folates and choline dietary intake are important in human reproduction, exerting a role in 1-C metabolism, which deliver methyl groups via the linked folate–methionine cycles for critical processes such as DNA synthesis, phospholipid, and protein biosynthesis [21,90]. Derangements in 1-C metabolism during the periconceptional period (in women from around 26 weeks prior to conception) are associated with reproductive failure and impact on implantation and long-term health [21,22]. The efficacy of the 1-C metabolism relies strongly on the folic acid cycle, especially on 5-methyl-tetrahydrofolate (5MTHF) as a methyl donor. *MTHFR* is the gene that encodes for methylenetetrahydrofolate reductase, the enzyme that converts folate to its biologically active form, 5-methyltetrahydrofolate [91]. This active form allows the re-methylation of homocysteine into methionine, using vitamin B12 as a cofactor [92], thus ensuring the correct functioning of the 1-C metabolism. However, variations in this gene cause reduction or loss of activity of the MTHFR enzyme leading to global hypomethylation and hyper-homocysteinaemia [21], a sensitive marker of deranged maternal 1-C metabolism, with important consequences also on fertility [93]. Various correlations, in fact, exist between high plasma homocysteine and adverse reproductive outcomes such as RPL [94], pre-eclampsia, placental abruption [95,96], and the prevalence of PCOS [97]. Moreover, a negative correlation is in place between derangements in 1-C metabolism, hyperhomocysteinaemia, and oocyte or embryo development, perhaps via aberrant methylation, but also via oxidative, vascular, apoptotic, and inflammatory pathways [98,99,100].

*C677T* (rs1801133) and *A1298C* (rs1801131) are the most common *MTHFR* gene variants. The carriers of these genotypes show reduced methylenetetrahydrofolate reductase activity compared to wildtype individuals, resulting in a higher risk of folate deficiency [91,92,101]. *C677T*, then, is common in Caucasians or Hispanics (about 20–40% are *CT*-heterozygous in USA and 8–20% are *TT*-homozygous in North America, Europe, and Australia), whereas Blacks are less affected by this genetic variant [91]. *A1298C*is mostly found in North Americans, Europeans, or Australians (7–12%) but less common among Hispanics and Asians (1–5%) [91]. Since *MTHFR* isoforms can impair gametogenesis and embryogenesis, genotyping both male and female partners for these mutations before conception might be considered, especially in the case of RPL and in gamete donors [102].

Since folate and choline pathways are coupled, folate intake might be modulated by the genetic variants involved in the metabolism of choline, specifically those relating to *PEMT* (rs7946 and rs12325817) and *MTHFD1* genes (rs2236225) [103]. SNPs in the former gene are associated with different risks of organ damage/dysfunction in the case of low dietary intake [93]. Within this context, identifying subjects with a reduction/loss of gene function associated with the metabolism of both folate and choline is a priority. In fact, an appropriate nutritional management based on their SNPs is advisable [101,102,104,105] (Table 1). For instance, food intervention studies showed that carriers for *C677T* had higher folic acid levels and reduced inflammation markers (ILs, TNF-α, and homocysteine levels) when exposed to high folate diets, especially from vegetables [106]. In patients carrying risk genotypes for both rs1801133 and rs1801131, instead, a natural folate-enriched diet should be recommended [107] together with adequate supplementation of the 5-MTHF active form according to their genetic profile (wildtype 200 μg/day, intermediate 400 μg/die and risk 800 μg/day) [101] (Table 1), thereby also avoiding the potential adverse effects of un-metabolized folic acid (UMFA) syndrome, which may occur when large doses of folic acid are used (5 mg/day) [102].

In addition, the adherence to MedDiet was associated with reduced homocysteine levels in carriers of risk alleles, both in homozygosity and heterozygosity, but not in wild types [128]. Furthermore, carriers of *MTHFR* mutations seem to benefit from an increased intake of choline, as carriers of *PEMT* and *MTHFD1* risk genotypes benefit from the increased amount of folate-rich foods [103,105,129].

Regarding supplementation, the B vitamin complex rather than only folic acid might be appropriate in specific cases (e.g., vegetarians or vegans) since vitamin B12 together with B6 act as substrates or cofactors in folate–methionine cycles. As a consequence, when intracellular B12 levels are low, the “methyl-folate trap” may occur, leading to a decline in intracellular folates [130].

In conclusion, since dietary inadequacies in B vitamins represent a growing problem in both developed and developing countries [131], testing *MTHFR* polymorphisms together with vitamin B status in the preconception period may be useful to highlight micronutrients deficiencies and to plan an adequate B vitamin-enriched diet (green raw vegetables, fruits, shellfish, etc.). Finally, it should always be considered that unhealthy habits such as smoking and excessive coffee and alcohol consumption deeply affect these pathways [132].

#### 2.1.2. Celiac Disease and Gluten Sensitivity

Celiac disease (CD) is a chronic inflammatory disease that affects approximately 1% of the world population [133]. It is characterized by gastrointestinal features, such as abdominal pain and distension, bloating, diarrhea, malabsorption, vomiting, and non-gastrointestinal features, such as chronic fatigue, iron deficiency anemia, dermatitis herpetiformis [134] after the ingestion of gliadin, a gluten protein, mainly present in wheat, but in barley and rye as well [135]. The causes of this disease are both genetic and environmental [136]. Genetic susceptibility is mainly imputable to human leukocyte antigen (HLA) haplotype DQ2 or DQ8, identified by molecular genetic testing of HLA-DQA1 and HLA-DQB1. Approximately 95% of patients with CD test positive for at least one of them [137], but their prevalence in the general population ranges from 30% to 40%, with only about 3% of carriers actually developing CD. In other terms, the absence of HLA-DQA1 and HLA-DQB1 excludes CD, while their presence simply defines an individual as susceptible [138]. Additional factors then play a crucial role in CD development (e.g., stressful events such as a loss a pregnancy, or an infection) [139].

In the case of CD, the ingestion of gluten triggers inflammation at the intestinal barrier and leads to the release of proinflammatory cytokines and autoantibodies, thereby inducing progressive atrophy of the intestinal villi, which causes nutrients malabsorption [140].Indeed, to establish a CD diagnosis beyond DQ2/DQ8 positivity, a positive celiac serologic test for tissue transglutaminase (tTG) IgA, anti-deamidated gliadin-related peptide IgA, and IgG and endomysial(EMA) antibody IgA together with specific histologic findings on small-bowel biopsy (partial or complete villous atrophy) are required [140].

CD has been associated with type 1 diabetes in children and adolescents, whereas different autoimmune endocrine diseases, such ashypothyroidism, hypoparathyroidism, hypopituitarism, or ovarian failure, were found to co-exist in adults [141]. Recent publications support that undiagnosed CD has a potentially negative impact on female reproduction [133,142,143,144] due to gynecological and obstetric disorders (e.g., delayed menarche, early menopause, amenorrhea) and/or adverse pregnancy outcomes (e.g., RPL, intrauterine growth restriction, low birth weight, and preterm deliveries) [143,145,146].

Specifically, CD may affect fertility through the malabsorption of micronutrients such as folic acid, fat-soluble vitamins, iron, zinc, and vitamin B12 [143], in turn leading to hyperhomocysteinemia and increasing the risk of thrombosis and coagulation alterations in general [147]. Other hypotheses involve immune-mediated mechanisms causing tissue damage and obstetric failures through altered placental function [55,144,145]. Endometrial inflammation during implantation was also suggested to be caused by a gliadin-linked aberrant expression of angiogenic and proinflammatory pathways [143].

Based on these data, when a patient with UI or RPL reports gastrointestinal and non-gastrointestinal issues, CD should be investigated [133,143,146,147,148,149,150,151] via a complete or partial screening (i.e., serological markers and/or genotyping for six SNPs in the *HLA* genes) (rs2187668, rs4713586, rs4639334, rs7454108, rs2395182, rs7775228) [133,143,145]. Nonetheless, intestinal and/or extraintestinal symptoms can occur in some patients even after the consumption of gluten-containing cereals, although both EMA and tTG antibodies are negative. There is, in fact, a condition called non-celiac gluten sensitivity (NCGS), whose prevalence ranges from 0.5% to 15% [152], that requires proper nutritional management [153] (Table 1).

In summary, the genetic predisposition for CD, along with an accurate anamnesis and serological tests, may unveil undiagnosed celiac patients or patients who may benefit from a gluten-free or gluten-reduced diet (3 g–13 g) [127] with a positive effect on their proinflammatory status and ultimately reproductive outcomes [141,142].

#### 2.1.3. BMI and Fat Mass

The impact of overweight and obesity on reproduction is mainly due to endocrine mechanisms that interfere with both ovarian and endometrial functions via altered inflammatory responses. The adipose tissue is an endocrine organ that secretes numerous bioactive cytokines, named adipokines, playing key roles in the regulation of immune response, glucose and lipid metabolism, and reproduction [154]. They include leptin, adiponectin, resistin, visfatin, omentin, and other non-adipose-specific cytokines such as IL-6, IL-1β, and TNFα. These molecules seem strongly associated with both insulin resistance (IR) and type 2 diabetes mellitus (T2DM), two well-known proinflammatory diseases [155,156].

The impact of obesity on ovarian function, granulosa cells, cumulus cells, and oocyte quality is subject to an intense investigation. Obesity is associated with lipid accumulation in non-adipose tissue cells, which increases oxidative stress and endoplasmic reticulum stress response, both phenomena tightly linked with systemic inflammation [62,157]. Likewise, the negative impact of obesity on endometrial receptivity has become evident: obese women show an increased risk of miscarriage [158,159,160,161,162] even when euploid embryos are transferredin utero [163,164,165]. However, the sole evaluation of BMI may lead to misclassification. Indeed, the concept of normal-weight obesity (NWO) has been outlined for normal-weight women with a proportion of fat mass (FM) greater than 30% [166,167,168,169]. Given the role of adipose tissue on inflammation, it is important to estimate FM in infertile patients through methods such as dual-energy X-ray absorptiometry (DEXA) and bioimpedance analysis (BIA) or using anthropometric measures in validated formulas to calculate the adipose mass [169,170].

Genome-wide association studies (GWAS) have identified several genetic variants associated with a higher susceptibility to obesity [171] when the subject is exposed to an obesogenic environment [10,172]. To date, over 300 polymorphisms have been identified. The main gene associated with FM and obesity is *FTO,* which regulates neurological and hormonal pathways, as reported in both mice and humans, associated with appetite and body energy consumption [173]. The most representative SNP in *FTO* is rs9939609 (*T*>*A*) which has been linked with BMI, T2DM, gestational diabetes (GDM), and eating behavior [174,175,176,177,178].

Based on the current evidence, in the case of *FTO* risk genotypes, hypocaloric MedDiets with low saturated fats and limited carbohydrates should be advised [108,109,110,111,112]. Especially patients carrying the risk alleles for rs1558902 may benefit from a high protein diet to lose weight [113]. Recently, the effects of the interaction between nutrigenetic variants and diet/lifestyle intervention on the mid-term changes in the anthropometric and clinical parameters of overweight or obese subjects affected by T2DM or dysglycemia have been evaluated [179]. This study showed that subject carriers of the *A* allele in *FTO* lost less weight and had a lower BMI decrease from baseline to 12 months than *TT* carriers, supporting the interaction between *FTO* and diet/lifestyle intervention in the regulation of body weight.

Further obesity-related genes include *MC4R*, peroxisome proliferator-activated receptor gamma (*PPAR-G*), and both adipokine-encoding genes (*LEP* and *ADIPOQ*) [180,181,182]. Carriers of *LEP* rs2167270 and rs7799039 show a higher risk of obesity and insulinresistance and should reduce the intake of carbohydrates, especially from sweets and snacks [114]. SNPs in the *ADIPOQ* gene encoding for adiponectin are associated with lower serological levels of the hormone, higher BMI, and elevated FM. Low SFA intake is recommended for *LEP* (rs2167270, rs7799039) and *ADIPOQ* (rs266729) risk genotypes [114,115]. Evidence suggests that, in patients carrying risk alleles, diets with an increased amount of both MUFAs and PUFAs, may reduce the risk of developing obesity [183,184,185] (Table 1).

#### 2.1.4. Milk, Dairy Products, and Lactose Intolerance

The main concern about dairy consumption in infertile patients is the contamination by steroid hormones, growth factors, pesticides, and chemical substances, often found in these products. All those substances might affect female endocrine functions and folliculogenesis [186]. Moreover, dairy consumption may increase the Homeostatic Model Assessment for Insulin Resistance (HOMA-IR) value, concurring with the risk of insulin resistance and PCOS [186]. However, the kind of product seems relevant. Indeed, the consumption of full-fat dairy products or fermented products, especially when added with probiotics and vitamin D, such as yogurt and kefir, seem beneficial in women with PCOS since they act on both tissue insulin sensitivity and glucose tolerance [186].On the contrary, skimmed milk intake seems associated with acne, a sign of PCOS, perhaps because androgen precursors are present in milk [186]. On the contrary, a recent meta-analysis suggests that a long-lasting consumption of low-fat dairy products is beneficial for tissue insulin sensitivity [187].

Given the contrasting results on a putative relationship between milk, dairy products, and fertility [16,17,186], there is no indication to remove them from the diet of an infertile patient [186,188]. Part of the discrepancies reported in the literature could depend on lactose digestion and absorption, which are highly variable across individuals. This variability depends primarily on genetics; however, conditions that affect the integrity of small-bowel mucosa might be involved as well (see Section 4) [189]. Lactose digestion and absorption depend on the enzyme lactase (encoded by the gene *LCT*) that hydrolyzes lactose into galactose and glucose in the small intestine [189]. This enzyme is essential during the first years of life, whereas its efficiency lowers with age [190]. In fact, lactose digestion in adulthood is the result of a status of tolerance defined “lactose persistence” [191]. Association studies demonstrated that lactose intolerance is due to hypolactasia with a genetic origin [190,192]: subjects at risk (*CC* genotype for *−13910C>T* (rs4988235)) are found mainly in the Caucasian Europeans [193,194]. *TT* genotype carriers, instead, show lactase persistence, while heterozygous *CT* genotype subjects show an intermediate condition and may present symptoms, as well [195]. The assessment of this variant might be useful from a nutritional standpoint. An *LCT*-deficient individual cannot digest lactose, and undigested lactose coming in contact with the intestinal microbiota undergoes fermentation causing visceral hypersensitivity, bloating, meteorism, and diarrhea [192], often associated with anxiety and the irritable bowel syndrome (IBS) [189]. The consequence is a proinflammatory status of the intestine and dysbiosis [196,197]. In these subjects, a diet low in lactose is strongly recommended. However, since this is not an allergy, a completely lactose-free diet is not required because patients with lactose intolerance often tolerate up to 12 g of lactose as a single dose with no or just minor symptoms [198]. Some forms of lactose maldigestion also improve following a targeted use of probiotics such as *Lactobacillus* spp., *Bifidobacterium longum,* or *Bifidobacterium animalis* [117,199] or lactase supplementation (3000 to 6000 IU of beta-gal [116] (Table 1). At last, genotype analysis is valuable for the differential diagnosis of primary and secondary hypolactasia to identify correct treatment through personalized dietary plans [200]. Importantly, commonly used tests such as the hydrogen breath tests (HBTs), i.e., the gold standard to measure the effective functionality of lactase in an individual [201], do not allow a differential diagnosis [189].

#### 2.1.5. Fatty Acids Metabolism

Among nutrients with inflammatory properties, PUFAs are probably the most important. PUFAs are fundamental building blocks of all cells, and their organization is essential for regulating cell functions [202]. Within the cell, membrane-derived fatty acids and their metabolites can regulate the antioxidant signaling pathway and modulate inflammatory processes, mainly via the inhibition of NF-kB and PPAR-alpha/gamma transcription factor pathways [203]. Omega-3 and omega-6 fatty acids are major PUFAs and are believed to be critical in the regulation of inflammatory and immune responses through pro- and anti-inflammatory activities, respectively. Omega-6 fatty acids and their derivatives (mainly arachidonic acid) are precursors of proinflammatory eicosanoids, whereas omega-3 and its derivatives (α-linolenic acid, eicosapentaenoic acid, and docosahexaenoic acid) are precursors of anti-inflammatory eicosanoids [204]. Moreover, several lipid mediators are biosynthesized from essential PUFAs (resolvins, protectins, and maresins). These molecules, known as specialized pro-resolving mediators (SPMs) [205], are involved in inflammation resolution [206]. In this view, they represent a novel promising therapeutic approach for PCOS [207,208], endometriosis [209], and some pregnancy-related pathologies [210]. Indeed, the pro-resolution strategy seems more promising than conventional anti-inflammatory approaches, at least for some conditions [211,212,213,214]. Different reports in the literature show a positive correlation between omega-3 PUFA and both spontaneous conceptions and IVF outcomes [17]. However, others failed to unveil such a correlation [215]. These contrasting results are imputable to different nutrient exposure and variable capacity to utilize/metabolize omega-3 among individuals. In this regard, the ratio between omega-3 and omega-6 amounts must be controlled since omega-6 PUFAs compete for the same enzymes involved in the omega-3 PUFAs pathway, making such a ratio crucial for the inflammatory balance [216]. The recommended value is 4:1 or less. Nonetheless, disproportionate amounts of omega-6 PUFAs are found in today’s Western diets leading to 10:1–50:1 ratio [216].

Not only the omega3-omega6 ratio can alter the inflammatory balance, but also genetic variants of the fatty acid desaturate genes (*FADS1*, *FADS2*, *FADS3*), as individuals carrying specific SNPs are more prone to a chronic proinflammatory status [217,218,219,220]. For example, *GG* carriers in rs174537 (*FADS1*) showed higher arachinoic acid, eicosadienoic acid (EDA), eicosapentanoic acid (EPA), low-density lipoproteins (LDL), and total cholesterol levels determining a higher proinflammatory status [218], and significant associations were also found for another *FADS1* polymorphism (rs174547) and decreased enzyme activity [221].Genotyping for these variants can help identify patients at risk for a chronic proinflammatory condition since the physiological resolution path is clearly underpowered in these individuals. However, an adequate amount of omega-3 PUFAs may compensate for an impaired enzyme function in individuals at risk, suggesting the importance of both tailored dietary plans [220,221] and tailored omega-3-PUFA supplementation [118] in individuals carrying risk alleles, especially if vegetarian [220,222,223] (Table 1).

#### 2.1.6. Glucose Metabolism

An unbalanced intake of macronutrients such as carbohydrates promotes both inflammation and oxidative stress. This occurs because simple carbohydrates (e.g., fructose and glucose) involve (i) de novo synthesis of free fatty acids (FFA) in the liver, causing lipotoxicity [224] and (ii) hyperinsulinemia, which leads to systemic inflammation [225] by stimulating NF-kB nuclear translocation, the extracellular release of proinflammatory mediators from macrophages and ultimately systemic insulin resistance [226]. In other terms, both lipotoxicity and hyperinsulinemia trigger inflammatory processes and increase ROS formation [155,227,228]. Increased intake of carbohydrates, in fact, has been associated with conditions such as obesity, metabolic syndrome, diabetes, leaky gut syndrome, and Alzheimer’s [229,230,231,232]. A large body of evidence supports the role of carbohydrates in fertility as well [233], and dietary adjustments to reduce insulin secretion represent an intriguing non-pharmacological perspective to counteract infertility. It has been demonstrated that, while LH and FSH are the primary regulators of late folliculogenesis, insulin can also modulate this process. Hyperinsulinemia, in fact, exerts both direct and indirect effects on folliculogenesis and intraovarian gonadotropin-driven granulosa and thecal cell steroidogenesis [234]. Moreover, high insulin levels and insulin resistance constitute an unfavorable biochemical environment in the ovaries [234,235]. Thus, reducing insulin circulating levels may reduce hormonal imbalance and improve ovarian function [236]. Insulin may also play homeostatic roles inenergy metabolism in the endometrium, with hyperinsulinemia contributing to poor implantation rates and increased miscarriage rates. The proposed mechanisms are (i) increase in androgens, plasminogen activator inhibitor, and uterine vascular resistance, (ii) decrease in glycodelin, insulin-like growth factor binding protein 1 (IGFBP 1), and uterus vascularity [237,238,239].

Genetic variants related to glucose metabolism were detected mainly in relation to T2DM. Several studies highlighted the complex polygenic nature of T2DM, where a plethora of genetic loci seems to increase the risk of its development by acting on insulin secretion or by reducing its action [239]. One of the most studied genes is peroxisome proliferator-activated receptor gamma (*PPAR-G*), which is part of the nuclear receptor subfamily of transcription factors involved in various biological processes that include the differentiation of adipocytes, lipogenesis, and glucose homeostasis [119,240]. In fact, some PPAR-G agonist drugs are used in the treatment of diabetes [241]. Among the SNPs analyzed, Pro12Ala (rs1801282) is the most relevant, whose minor allele is *G* [240]. As evidenced by a recent meta-analysis, carriers of *GG* genotype seem less subject to T2DM [240]. The protective role of this SNP seems confirmed in insulin resistance, and screening for this SNP would identify individuals exposed to lower risk, thus guiding the professional in establishing the daily percentages of carbohydrates in the diet. Furthermore, both *GG* and *GC* genotypes are subject to a greater weight loss than *CC*, especially when they consume a diet rich in MUFAs, thus emphasizing the importance of the type of fatty acids chosen for them (e.g., extra virgin olive oil) [119,120]. Another relevant gene is transcription factor 7-like 2 (*TCF7L2*), which encodes for a transcription factor central in the WNT signaling pathway. This transcription factor covers, among other functions, an essential role in diabetes by mediating the expression of glucagon-like peptide 1 (GLP-1) [242,243]. *TCF7L2* variants, including mainly rs12255372 and rs7903146, predict the prevalence of T2DM in high-risk individuals, suggesting putative synergistic effects between different risk factors [244]. *TT* carriers for rs12255372 were associated with increased T2DM prevalence, especially in the case of diets with high glycemic index and load, thus suggesting low carb and low GI diets in patients at risk [121]. Another *TCF7L2* variant (rs7903146) is associated with higher blood sugar levels and impaired insulin response [245]. Subjects at risk have significant benefits from following Mediterranean diets, characterized by high fiber intake, and should be preferred in *TCF7L2* risk allele carriers [122,246]. T2DM-related gene variants have also been associated with GDM susceptibility, which represents the most common metabolic disorder of pregnancy [177,178]. In this contest, Franzago et al. showed an increased risk of GDM in women carriers of the *TT* genotype of the *TCF7L2* rs7903146 [177,178], as well as an association between nutrigenetic variants in *PPARG2*, *APOA5*, *MC4R, LDLR,* and *FTO* genes and lipid parameters at the third trimester of pregnancy. Since women with GDM are at greater risk of cardiovascular disease (CVD), T2DM and metabolic syndrome later in life, these findings could allow the development of an easy tool for personalized intervention strategies, including routine anthropometric and biochemical parameters, dietary assessments, and genetic make-up [247].

Another relevant gene involved in insulin response is potassium inwardly rectifying channel subfamily J member 11 (*KCNJ11*). KATP channels for potassium, which are expressed in pancreatic β-cells with a role in insulin secretion, are altered in case of mutations in this gene, thereby leading to hyperglycemia [248]. A missense mutation, rs5219 *C*/*T*, was associated with an increased risk of T2DM in Caucasian and East Asian populations, and the effect of this SNP is amplified by high BMI [248,249].

In summary, genotyping for *PPAR-G*, *TCF7L2* and *KCNJ11* might help clinicians establish personalized nutritional management in subjects who must undertake a low-glycemic-index diet with the characteristics of the MedDiet with adequate fiber intake (30 g/day), limitation of refined carbohydrates, and replacement of animal fats with vegetable ones, especially MUFAs and PUFAs (e.g., oily fruit) [119,120,121,122,123,124,125] (Table 1). Genotyping would also allow calculating the risk of T2DM later in life and the risk for gestational diabetes, whose short and long-term consequences for the mother, the fetus, and the offspring are not negligible [250]. Of course, again, the interaction between genetics and the environment is crucial [251]. For nutritional strategies aimed at managing blood glucose homeostasis refer to Section 5.3.

#### 2.1.7. Caffeine’s Metabolism

Caffeine (1,3,7-trimethylxanthine) is the worldwide most consumed stimulant, whose main sources are coffee, tea, soft drinks, and chocolate [252]. Its consumption may modulate embryo implantation and early post-implantation behavior [252,253,254] and decrease live birth rates in both spontaneous conception and IVF [255]. Additionally, several studies linked caffeine intake to a longer time of pregnancy with a possible dose–response effect [256,257,258,259] and increased risk for fetal death and stillbirth [253]. At present, there is little evidence to support the detrimental effect of mild–moderate caffeine consumption on fertility and IVF outcomes [257,260]; therefore, the complete abstinence before or during pregnancy cannot be supported [258]. However, coffee and energy drinks containing caffeine can induce neural and vascular changes with pro-aggregatory effects and a consequently higher risk of thrombosis [261,262]. However, it is complex to assess a putative inflammatory response to caffeine. Data suggest a predominant anti-inflammatory action of coffee probably due to components other than caffeine, such as trigonelin and chlorogenic acid [263]. The current recommendation for pregnant women or women attempting to conceive is to limit caffeine intake to 200 mg/day, i.e., 1–2 cups [255,258,260]. These recommendations, though, represent a general guideline for the population that does not consider an individual’s genotype.

A total of 95% of caffeine is metabolized by cytochrome P450 1A2 (CYP1A2) in the liver, which exhibits great individual enzyme variability [264]. Numerous studies showed that different individuals have different degradation capacities and that the risk associated with caffeine consumption, for example, myocardial infarction, varies according to the enzymatic capacity dictated by the genetic profile [265]. A missense mutation of this gene (rs762551) reduces its enzymatic activity, and consequently, carriers of the *C-*allele, both in homozygosity and heterozygosity, are considered “slow metabolizers”. In fact, in these subjects, blood pressure increases significantly with caffeine consumption [264]. Instead, AA genotype carriers are “fast metabolizers”, which makes them less sensitive to this molecule. Given the known role of variants in the *CYP1A2* gene in affecting caffeine metabolism, the evaluation of circulating caffeine levels and its metabolites (e.g., serum paraxanthine) together with targeted genotyping might be important [260]. In subjects at risk, classified as “slow metabolizers”, especially in homozygosity, the recommendation not to exceed 200 mg per day may not be sufficient, and in dietary counseling, genotyping maybe be useful to sensitize a patient to further reduce the intake of caffeine to 100 mg per day, especially during ART treatments and in the first phase of pregnancy [126] (Table 1).

## 3. Nutrigenomics and the Management of Low-Grade Inflammation in Infertile Patients

Nutrients should not be considered a mere source of energy. They also contribute to the regulation of gene expression, either directly or via reversible and heritable epigenetic changes [81]. The branch studying the influence of nutrients on DNA is called nutrigenomics [26]. In general, there are three different mechanisms of action through which epigenetic changes occur [266]. These include DNA methylation (gene silencing), histone modifications, mainly through acetylation (gene activation via increased access to chromatin), and post-transcriptional modifications through RNA-dependent mechanisms [266,267]. Mounting evidence suggests that epigenetic alterations may impact inflammatory processes, thus contributing to the development of pathologies such as diabetes, cardiovascular diseases, cancer, and neurological disorders [268]. Of note, epigenetic alterations often occur in infertile patients [269,270,271]. Since nutrients are one of the most important epigenetic modulators, it would be interesting to understand the relative role of each nutrient in potential epigenetic alterations. The following section will be dedicated to some key foods and their bioactive compounds, with proven epigenetic effects on inflammatory pathways and, therefore, useful in the nutritional management of infertile patients for lowering their proinflammatory status 

### 3.1. Folates

Folates represent the most evident example of how diet can modulate gene expression since methylation is a key mechanism of epigenetic and imprinting processes [22,272]. Indeed, vitamin B9 is the key methyl donor, able to trigger a cascade of events such as DNA and histone methylation that activate or repress specific genes [267,273]. Methylation errors, deriving from either hypo- or hypermethylation, are due to folate deficiency and have been largely associated with major health problems such as cancer, metabolic, autoimmune, and mental disorders [270,272]. However, the most investigated area in relation to folate intake is early embryonic development [267]. Low folate intake during the perinatal period has been associated with permanent hypomethylation and incorrect gene expression, which are transmitted to future generations [93,267]. Impaired methylation may also impact reproductive health. Several studies have shown that gamete quality, ovulation, corpus luteum formation, and embryo development might all be impaired [270]. The negative effects are mostly attributable to oxidative stress [270]. Methylation and oxidative stress are linked by the 1-C cycle, which stimulates the synthesis of glutathione and recycles homocysteine [270]. Glutathione, a powerful antioxidant, protects gametes from reactive oxygen species (ROS) [270]. Folate and all B vitamins involved are direct supporters of the 1-C cycle and contribute to maintaining low oxidative stress, thus avoiding epigenetic alterations [270]. Furthermore, folate deficiency is directly related to increased levels of inflammatory markers such as IL-β, IL-6, and TNF-α [274]. In some studies, high doses of folate have been used for their anti-inflammation properties [275]. A correct intake of folate also reduces the risk of obesity in genetically predisposed individuals, and the risk of altered glucose metabolism and hypertension, via epigenetic pathways [276,277,278]. Therefore, adequate amounts of B vitamins, mainly from foods rich in folate and vitamin B12, are essential to guarantee correct epigenetic pathways through the synthesis of methionine and S-adenosyl methionine (SAM). These vitamins are important, especially during ART in advanced reproductive age, since both these techniques and aging may exacerbate methylation defects [270]. In general, due to dietary inadequacies of B vitamins in both developed and developing countries [131], supplementation is mandatory prior to conception and during pregnancy to prevent defective methylation in key genes [21].

### 3.2. Dietary Fatty Acids

Fatty acids exert different epigenetic functions depending on their category [279]. Saturated fatty acids (SFA) display well-known harmful effects through modulation of DNA methylation and histone acetylation, ultimately leading to inflammation, lipotoxicity, and metabolic alterations [279]. Indeed, diets rich in SFA, such as the Western diet, contributed to a higher prevalence of obesity, dyslipidemia, diabetes, and cancer [279]. Nevertheless, not all SFA are associated with negative consequences. This is the case of short-chain fatty acids (SCFA), deriving from intestinal microbial fermentation of indigestible foods, which were linked to positive outcomes on carcinogenesis and inflammation via epigenetic pathways [280]. Sodium butyrate, for example, arrests cell proliferation and promotes apoptosis by inhibiting HDAC activity and inducing histone hyperacetylation of specific genes involved in colon carcinogenesis [280]. The positive role of butyrate was also confirmed on metabolic biomarkers of inflammation by mediating transcription factors such as NF-kB and inhibition of HDAC [281]. Several clinical trials reported lower plasma *C-*reactive protein (CRP) levels and decreased inflammatory markers (e.g., intestinal fecal calprotectin) when consuming fiber-enriched diets associated with higher colonic production [281]. Dietary trans-fatty acids (TFAs), then normally present in industrial foods (e.g., margarine, crackers, bakery products, and deep-fried foods) [282], also exert proinflammatory properties [283]. If these products are consumed on a regular basis, they may result in a >3-fold increase in plasma concentrations of CRP and significantly increased concentrations of TNFα, chemokine (C-C motif) ligand 2 (CCL2), IL-1β, and IL-6 [284,285,286,287]. Furthermore, TFA consumption is associated with a higher risk of ovulatory infertility, endometriosis [17], and insulin resistance [234,288]. In addition, TFAs seem to impact global DNA methylation [289] as well as the regulation of HDL plasma lipids [279].

Even unsaturated fatty acids, such as PUFAs, may act on DNA epigenetic modulation [279]. Indeed, long-chain fatty acids such as omega-3 and omega-6 are rich in phospholipids that figure among the major methyl group acceptors in the 1-C metabolic pathway [290]. However, they seem to induce distinct epigenetic changes in adipose tissue accumulation, obesity, FA uptake and transportation, insulin resistance, and inflammation. PUFAs overfeeding changes the methylation of 1797 genes in human adipose tissue, whereas SFA overfeeding increases the methylation of 125 genes, with just 47 genes modified by both regimens [291]. Omega-3 lipids exert epigenetic modulatory effects on gene expression favoring an anti-inflammatory status, leading to a significant upregulation of the genes encoding for the PPAR-G and a downregulation of the genes encoding for the low-density lipoprotein (LDL) receptor and interleukin-1 [292]. Omega-3 PUFAs modulate inflammation even through enhanced expression of DNA methyltransferases (DNMTs) and increased LKB1 tumor suppressor gene expression, in turn stimulating LKB1 activity with a consequent inhibition of glycolytic enzymes and targeting rapamycin (mTOR) signaling [279,293]. Omega-6 lipids, instead, promote a proinflammatory status, affecting the concentrations of proinflammatory cytokines through DNA methylation of TNFα [294].

MUFAs induce hypomethylation, with positive consequences on inflammation [279]. OA is undoubtedly the most important MUFA, whose consumption ameliorates lipid and inflammatory profiles and reduces the risk of cardiovascular, metabolic, and neurodegenerative diseases even via epigenetic mechanisms [279,295,296,297]. Its main dietary source is extra virgin olive oil (EVOO), key in the MedDiet. Besides OA, EVOO contains other valuable components such as phenolics, phytosterols, tocopherols, squalene, vitamins E and K, all exerting both anti-inflammatory and antioxidant effects, modulating glucose metabolism, and moderating endothelial dysfunction [297,298,299,300]. The synergy between its components makes it very efficient in modulating risk factors for various diseases [297]. Nutrigenomic studies on EVOO showed that its positive properties are exerted by acting on both the transcriptome and the miRNome [297]. The consumption of EVOO was linked to the downregulation of the CD40/CD40 ligand, a member of the TNF-family involved in immune responses, and to a reduced expression of proinflammatory cytokines with an impact on inflammation-related genes, such as *IFN-y*, *IL-7R*, and *IL8RA* [297]. EVOO decreases plasma LDL oxidation and modulates blood pressure by acting on the renin-angiotensin-aldosterone system (RAAS). Post-prandial studies investigating mRNA expression report a positive effect on disorders such as the metabolic syndrome [297].

These data overall confirm the importance of diet in the regulation of cellular metabolism and suggest lowering the intake of SFA and TFA while increasing omega-3 PUFAs and MUFAs so to decrease both inflammation and oxidative stress in infertile patients via epigenetic mechanisms.

### 3.3. Spices: Turmeric, Ginger, and Chili Pepper

For centuries, plants, including spices, have been used to treat several chronic diseases [301]. In fact, roots, leaves, seeds, or berries, generally called spices, have healthy properties [301]. The nutraceuticals derived from spices have been largely studied in the prevention and treatment of inflammatory states [301]. Hereafter we focused on turmeric, ginger, and chili pepper, the main spices with proven antioxidant and anti-inflammatory properties, which could play a crucial role in ameliorating the inflammatory process in infertile patients [302,303,304].

#### 3.3.1. Turmeric

Turmeric or *curcuma longa* is part of the Zingiberacee family and is a plant whose roots are rich in molecules with nutraceutical properties, including curcumin [302]. Curcumin is a polyphenolic compound with anti-inflammatory, antioxidant, and anti-lipidemic activities. Curcumin is an epigenetic inactivator for genes involved in neurodegenerative and chronic diseases, including cancer [305]. The epigenetic mechanisms are related to DNA methylation, histone modification, and miRNA modulation, but also to the activation of transcription factors, cytokines, chemokines, and the inhibition of angiogenesis through apoptotic mechanisms [306]. This pleiotropic molecule is also involved in inflammatory processes, stimulating an anti-inflammatory response in both acute and chronic phases [305]. These properties make the compound a candidate for anti-inflammatory treatments [307]. In Middle Eastern cuisine, curcumin is widely used in cooking, combined with pepper, which increases its absorption. Scientific evidence about its properties has helped export this spice to other countries as well [302].

#### 3.3.2. Ginger

*Zingiber officinale*, commonly called ginger, is also a herbaceous plant belonging to the Zingiberacee family whose rhizome possesses essential oils with antioxidant, anti-inflammatory, antimicrobial, and anti-glycant properties [303]. The use of this plant in phytotherapy started in Asia, mainly for the treatment of gastrointestinal disorders, from nausea and dyspepsia to the irritable bowel disease or the infection from Helicobacter pylori [308]. Its anti-inflammatory properties have been attributed to the inhibitory role exerted on the synthesis of prostaglandins and leukotrienes [303,309]. A clinical trial conducted on endurance runners who consumed 500 mg of ginger powder after exercise registered significantly lower levels of inflammatory markers, specifically plasma cytokines [310]. Furthermore, ginger is currently used to treat cases of heavy menstrual bleeding [311].

#### 3.3.3. Chili Pepper

Capsaicin is part of the capsaicinoid family, and it is a component of chili pepper [304]. This molecule has been investigated in interventional studies for its antioxidant and analgesic, as well as anticancer properties [312]. In addition, its effects have been studied in obesity. Capsaicin acts on metabolism, increasing thermogenesis and contributing to the reduction in fat, especially visceral fat [313,314]. Furthermore, capsinoids, the secondary metabolites of capsaicin, increase the feeling of satiety [314,315]. Capsaicin can contribute to the treatment of pain not only for its analgesic properties but also for inhibiting the expression of inflammatory cytokines, thus counteracting the effects of many chronic inflammatory and autoimmune diseases [312,316].

## 4. Microbiomics in the Nutritional Management of the Infertile Patient

NGS and other high-throughput technologies allowed recent advances not only in genomics but also in microbiomics, namely the study of the microbiome, that is, the totality of microbes in specific environments (e.g., the human gut) [24]. The Human Microbiome Project (HMP), analyzing the genetic material recovered from distinct sites on the human body, has highlighted the physiological microbial abundance of multiple strains and species of different phyla in different sites of the human body, above all gut [317,318]. The human digestive tract is considered an endocrine-metabolic organ, and it hosts a symbiotic microbial community [319,320]. Most of the microorganisms found in the digestive tract belong to groups of Firmicutes, Bacteroidetes, Proteobacteria, and Actinobacteria [321]. A normal balance between Bacteriodetes and Firmicutes is mandatory to maintain intestinal homeostasis, whereas a higher Bacteriodetes/Firmicutes ratio indicates dysbiosis [322]. Gut microbiota (GM) activity and diversity, determining the state of “eubiosis”, affect human health [323]. Eubiosis is crucial for intestinal barrier integrity, in turn essential to preventing the permeation of antigens, endotoxins, pathogens, and other proinflammatory substances in the human body. It also contributes to energy balance, the synthesis and absorption of nutrients (including vitamins and short-chain free fatty acids), the metabolism of glucose, lipid, and bile acids, and the feeling of satiety [324,325,326,327]. Furthermore, it is involved in local and systemic modulation of the immune and inflammatory response [281,319,324,325,328]. Dysbiosis, namely a condition that occurs when microbiota deviates from the “eubiotic” or “healthy” status, can lead to an alteration of the intercellular tight junctions responsible for the integrity of intestinal mucosa and its permeability, thus causing the leaky gut syndrome (LGS) [329]. LGS induces chronic low-grade inflammation, both because the mucus layer becomes more permeable to microbes and microbial products and because of the activation of the mucosal-associated lymphatic tissue (MALT), which in turn triggers the inflammatory cascade (leukocytes, cytokines, TNF-α) and results in tissue damage [330]. Indeed, persistent gut dysbiosis is strictly related to inflammatory bowel diseases such as ulcerative colitis, Crohn’s disease, and indeterminate colitis [331,332]. Emerging evidence indicates that the composition of the GM affects fertility. Indeed, GM can modulate circulating concentrations of sex hormones such as estrogens, testosterone, progesterone, and corticosteroids [333]. GM and estrogens show the tightest correlation [334]. GM impacts estrogen concentrations in the host through the secretion of β-glucuronidase, which deconjugates estrogens, enabling them to bind estrogen receptors with the subsequent physiological consequences downstream [335,336]. GM mainly acts on estrogen metabolism by modulating the enterohepatic circulation of estrogens. Therefore, a woman’s GM may, in part, reflect the metabolic functioning of her hormonal balance and, therefore, her reproductive health [333].

GM dysbiosis worsens PCOS and insulin resistance conditions [324,337,338,339,340,341], and it contributes to the onset and progression of endometriosis [325,342]. GM biodiversity is decreased in PCOS women, showing elevated Escherichia:Shigella ratio and an excess of Bacteroides compared to healthy women [341]. In the case of endometriosis, instead, *Lactobacillus* concentrations and potentially pathogenic GM profiles (altered Firmicutes:Bacteroidetes ratio) were highlighted [325]. Immune system dysregulation and altered estrogen metabolism are two conditions also triggered by a dysbiotic status involved in the pathogenesis of endometriosis [325]. An association exists even between obesity and dysbiosis, with increased Firmicutes:Bacteroidetes ratios [322] that result in LGS. In this syndrome, an excess of lipopolysaccharides (LPS) enters the bloodstream, thus contributing to systemic inflammation and metabolic alterations. The mechanism is mediated by the binding to toll-like receptor 4 (TLR-4) in the intestinal epithelial, in turn resulting in cytokine production, including TNFα and IL-6 [62]. Thyroid autoimmune diseases, often a comorbidity of infertility, are associated with GM [343,344,345]. Similarly, increased ratios of *Butyricimonas*, *Dorea*, *Lachnobacterium,* and *Sutterella*, have been reported in women with POI compared to healthy controls [346]. Lastly, GM influences both vaginal and uterine environments due to their continuity with the gut [347]. In fact, Lactobacilli (mainly *L. crispatus*, *L. gasseri*, *L. iners,* and *L. jensenii*), i.e., the main components of a healthy vaginal microbiota, come from the gut [348]. The integrity of both vaginal and uterine microbiota is critical for a healthy reproductive system. Specifically, the vaginal microbiota protects uterine health [349], while uterine microbiota seems responsible for endometrial receptivity [350,351].

Based on all this evidence, assessing GM composition should be considered an important diagnostic tool in infertile patients, especially since it is actionable via dietary strategies to restore and/or maintain gut eubiosis through targeted nutrition [352,353,354,355] or probiotic supplementation [356].

### 4.1. Possible Test to Assess Gut Integrity and Microbiota Composition

Fecal sample analysis is the simplest way to investigate gut integrity and microbiota composition via either metabolomics or genomics. Among the metabolites, calprotectin and zonulin are of particular importance. Zonulin is a protein involved in the regulation of paracellular transport in the intestinal lumen. Increased zonulin concentrations correlate with increased intestinal permeability [357,358], a condition associated with inflammatory (e.g., T2DM, CD, obesity) and autoimmune diseases (Crohn’s disease) [359,360,361]. Calprotectin, instead, is a protein secreted from stimulated neutrophils, eosinophils, and monocytes, and when found in feces, it is considered a marker for inflammatory bowel diseases (e.g., Crohn’s disease and ulcerative colitis) [362,363]. Other fecal metabolites are described elsewhere [364].

Microbial compositions can also be assessed in fecal samples through 16S rRNA gene amplicon sequencing [365]. This type of analysis allows a phylogenetic microbiota profiling for taxonomy composition, which may translate into clinically useful information for differential diagnosis [366] and therapeutic strategies, such as appropriate nutrition and targeted probiotic supplementation aimed at restoring gut homeostasis [356].

Also, urinary metabolites have been proposed to assess dysbiosis, mainly indican (3-indoxyl sulfate) and 3-methyl-indole (also named skatole). Those are two tryptophan catabolites found in traces in urine when microbial metabolism is altered [367]. While increased indican concentrations indicate a fermentative dysbiosis in the small intestine [368], increased urinary skatole is indicative of colon inflammation [369].

### 4.2. The Role of Diet to Ameliorate Gut Microbiome and Intestinal Barrier Function

Dietary intervention may alter the composition and activity of GM [370]. A calorie-balanced MedDiet, high in vegetables, PUFAs, dietary fiber, and low in simple sugars and saturated fatty acids, is beneficial to developing and maintaining the microflora [321,326,328,371]. Additionally, supplementation with natural anti-inflammatory and anti-oxidative substances (e.g., fermented plant foods, *curcuma longa*, coenzyme Q10, zinc) and multi-strain probiotics (e.g., *Bifidobacterium* and *Lactobacillus*, normally found in fermented foods such as yogurt and kefir), prebiotics (e.g., fructooligosaccharides, inulin, and galactooligosaccharides) and synbiotics, can contribute to the healthy composition of the gut microbiota and, thus, improve fertility-related and pregnancy-related disorders (e.g., PCOS, endometriosis and gestational diabetes mellitus) [321,324,325,328,339,372,373,374,375]. A low-carbohydrate diet has also been proposed in case of leaky gut disfunction and PCOS [339,372]. However, the central principles of a “leaky gut diet” are low carbohydrates, no/low milk and dairy products, and no/low gluten, as they all represent the main factors triggering LPS-induced immune-inflammatory response [376].

#### 4.2.1. Gluten-Free Diet

A gluten-free diet (GFD) has been proposed for the leaky gut diet to attenuate intestinal barrier dysfunction and inflammation [372]. A GFD is strictly recommended for celiac patients; however, it has some limitations [377]. In fact, celiac patients tend to compensate for gluten absence with improper eating habits, which are high in saturated fats, hypercaloric drinks, sweets, salty snacks, and highglycemic grain products, and low in dietary fiber and nutrients [377]. In addition to the improperly balanced GFD, the use of highly processed gluten-free products can further negatively affect celiac health [377]. In fact, many gluten-free foods are characterized by an elevated glycemic index, a high content of saturated fats and salt, and a reduced amount of minerals and vitamins [378,379]. Thus, a balanced GFD with whole meal GF cereals with high nutritive value (e.g., quinoa, buckwheat, teff, and amaranth) should be recommended [378]. Furthermore, some authors observed differences in the gut microbiota composition (involving mainly *Lactobacillus* and *Bifidobacterium* strains) in people treated with the GFD [377,380], suggesting cause-effect relationships with dysbiosis [377]. For all these reasons, a structured GFD should be administrated only when strictly necessary [381] and should be focused on the consumption of naturally gluten-free foods, such as quinoa, rice, buckwheat, sorghum, tapioca, millet, amaranth, teff, etcetera [382].

#### 4.2.2. The Controversial Role of Milk and Dairy Consumption

To reduce the immune-inflammatory response, the leaky gut diet also suggests a diet free of milk and dairy products [372]. Undigested lactose can induce osmotic load, and its subsequent fermentation by the GM induces inflammatory processes in the mucosal surface [383,384]. When lactose indigestion is not due to hypolactasia of genetic origin (see Section 2.1.4) but is related to an abnormal condition of the intestinal barrier (leaky gut and/or dysbiosis), it is possible to act with proper dietary intervention. In these patients, the avoidance of dairy food is not resolutive but can only mitigate their symptoms. In these cases, it is better to restore their barrier functionality and a physiologic GM. The treatment includes gluten and lactose-reduced diets, paying particular attention to alternative sources of calcium, enzyme replacement, and type of food [384,385]. Indeed, certain types of foods, although derived from milk, seem even beneficial for GM (e.g., yogurt, kefir). In fact, fermented foods, well tolerated also in case of lactose intolerance [385,386,387], contain relatively stable microbial ecosystems, composed primarily of lactic acid bacteria (LAB—among the others, *Saccharomyces* yeasts and *Bifidobacteria* spp.), and LAB primary metabolites (e.g., lactic acid), which are considered probiotic and help to overcomegut dysbiosis [388,389]. Ghee “clarified” butter is also an important source of butyric acid, or butyrate, a short-chain fatty acid with well-known nutraceutical anti-inflammatory properties because it increases concentrations of tight junction proteins and improves intestinal permeability [390,391].

At last, it should be noted that lactose intolerance is not harmful in these patients, provided it does not exceed the fermentative capacity of the gut flora [385]. For these reasons, the complete avoidance of dairy foods is no longer necessary, whereas finely regulated dietary quantity and quality are advisable [386,387,392].

### 4.3. Alcohol and Gut-Associated Inflammation

Many studies reported a link between alcohol and adverse consequences on fertility, such as reduced fertilization rates [393], increased risk of miscarriage [394], premature birth, low birth weight, and fetal alcohol spectrum disorder, including fetal alcohol syndrome [258]. The mechanisms by which alcohol affects female fertility include the HPG axis dysregulation [395]. Indeed, alcohol alters estrogen and progesterone levels suppressing folliculogenisis, causing anovulation and luteal phase dysfunction [253,258]. Furthermore, alcohol is a potent calorigenic agent, also unfitted for the control and maintenance of normal weight [259]. However, another important mechanism by which alcohol may interfere with fertility is by inducing intestinal inflammation. Indeed, it has been demonstrated that alcohol and its metabolites lead to intestinal inflammation, increasing the permeability of the intestinal lining, altering intestinal microbiota composition and function as well, thus, affecting the intestinal immune homeostasis [396,397]. Therefore, alcohol consumption should be avoided not only during pregnancy for its proven negative effects [398] but also during the periconceptional period for women seeking a pregnancy since no recommended safe limit of alcohol intake exists [399].

### 4.4. The Key Role of Vitamin D for Intestinal Homeostasis

Vitamin D (cholecalciferol) is pivotal for intestinal homeostasis, ensuring appropriate levels of antimicrobial peptides in the mucus, maintaining epithelial integrity by strengthening intercellular junctions, and thus preventing LGS [400]. Indeed, vitamin D deficiencies are responsible for altered integrity of the gut epithelial barrier. Furthermore, vitamin D, by binding its receptor VDR, influences both the innate and the adaptive immune systems representing an important suppressor of the inflammatory response [400]. Indeed, vitamin D can (i) regulate the expression of genes generating proinflammatory mediators, (ii) interfere with transcription factors, such as NF-kB, that regulate the expression of inflammatory genes, and (iii) activate the signaling cascades, such as MAP kinases, which mediate inflammatory responses [401].

A large body of evidence suggests that vitamin D is involved in the modulation of women’s fertility. VDR is found in reproductive tissues such as the ovary, uterus, placenta, pituitary, and hypothalamus [402]. Moreover, vitamin D might have beneficial effects on metabolic/hormonal parameters of PCOS and endometriosis and perhaps IVF outcomes [403]. However, no cause-effect relationship has yet been established. Probably, altered vitamin D concentrations, rather than exerting a direct role on fertility, is a sign of defective intestinal homeostasis, with subsequent consequences on immune balance in patients [400]. Therefore it is of utmost importance either to adopt nutritional intervention to ensure a correct intake of vitamin D (e.g., consumption of salmon, mushrooms, sardines, eggs) [404], eventually also using vitamin D supplements, or to investigate the integrity of the intestinal barrier of the patient [405] in order to ensure correct intestinal homeostasis.

## 5. Other Strategies Aimed at Managing Chronic Low-Grade Inflammation in Infertile Patients

### 5.1. Dietary Caloric Restriction (Intermittent Fasting) and Its Anti-Inflammatory Properties

Dietary caloric restriction (CR), without severe nutritional deprivation, has been shown to exert an anti-inflammatory effect by modulating mitochondrial metabolism and autophagic flux, protecting the intestinal barrier, dampening inflammation, and inhibiting the transcription of critical genes such as NF-kB [406]. Increasing evidence suggests potential benefits from intermittent fasting and caloric restriction on markers of health and longevity [407,408]. During CR, there is a decline in glycolytic rates in favor of respiratory metabolism as the main energy source. These alterations change the equilibrium of the reduced/oxidized forms of NAD toward NAD+, which works as a coenzyme in many biological processes and energy production. NAD+ is absolutely required in the reaction catalyzed by sirtuins, class III histone deacetylases that act as energy sensors [409]. Thus, CR is believed to reduce inflammation and aging by boosting the activity of some sirtuins. Activities of sirtuins toward several transcription factors and cytoplasmic protein substrates, beyond histone deacetylation, make them master regulators of cellular homeostasis, oxidative stress, inflammation, metabolism, and senescence. Although SIRT1, the most studied member of the sirtuin family, plays an important regulatory role in reproductive physiology [410], positive and negative effects of CR on female reproduction have been demonstrated. In mice, CR increases reproductive capacity and prolongs fertility lifespan. In humans, CR results in hypothalamic amenorrhea but exerts beneficial effects on PCOS in obese women [411].

Since CR seems to act mainly by enhancement of mitochondrial function, and the key role of mitochondria for oocyte and embryo competence is well known [412,413,414], this strategy deserves future investigations also in relation to reproductive processes.

### 5.2. Ways of Cooking and Advanced Glycation End-Products

The method of cooking foods dramatically affects inflammatory processes. When cooking with high heat under dry conditions, such as grilling, or during thermal processing of foods such as the ones occurring in industrial food production, dangerous compounds named advanced glycation end-products (AGEs) may form [415]. AGEs are also formed endogenously as by-products of metabolic processes. With their prolonged half-life, they gradually accumulate under oxidative stress and inflammation, becoming implicated in aging-related dysfunctions, diabetic complications, and pathogenesis of numerous diseases [416], including female reproductive dysfunctions [417]. AGEs appear to affect cell function through two main mechanisms: (i) crosslinking proteins, directly altering their structure and function; (ii) activating different cell receptor-mediated and receptor-independent mechanisms, which lead to increased oxidative stress and release of proinflammatory cytokines [418]. Moreover, the majority of food AGEs escape digestion and absorption, ending up directly in the colon, where they seem to modify local microbiota metabolism and modulate gut integrity and inflammation [419]. Unfortunately, western-type dietary patterns include ultra-processed foods and refined carbohydrates [420], and even if some “healthy dietary pattern” exists, such as the MedDiet, which is mainly based on the consumption of vegetables, fruits, cereals, nuts, and legumes, most of them are cooked by adding substantial amounts of olive oil. When “high-AGE” ways of cooking are applied (e.g., avoiding olive oil for caloric issues), the anti-inflammatory effect can be partially lost. As a result, a certain food of the same caloric content but exposed to different cooking methods during the same period would have very different AGE content. Therefore, since AGEs represent exogenous boosters of inflammation, it is important to guide patients on the correct cooking technique. In the last decade, the negative impact of AGEs on female fertility has gained a great deal of attention. From current literature emerges that altered AGE deposition represents a common feature in all PCOS phenotypes [417]. Moreover, intraovarian AGE represents key factors in the vicious circle centered on oxidative stress underlying reproductive aging [421]. Therefore, in addition to applying correct cooking, nutritional strategies to limit AGE-damage focus on patterns, foods, and compounds that limit hyperglycemia, a diet rich in antioxidants and anti-inflammatory foods and antiglycation medicinal plants are advised [422].

### 5.3. Nutritional Management of Blood Glucose Homeostasis

The proinflammatory and pro-oxidant effect of unbalanced dietary carbohydrates, acting via both lipotoxicity and hyperinsulinemia, has been discussed in Section 2.1.6. Appropriate dietary interventions for the management of blood glucose include reduced intake of simple sugars, especially from sweet drinks, sweets, and fruit juice, in favor of complex, low-glycemic-index carbohydrates (e.g., whole grain or ancient grains rich in dietary fiber) [423,424]. Particular attention should be given to the management of the postprandial glycemic and insulinemic response. Therefore, it is not only important to consider the glycemic index of single foods included in a meal, but also to calculate the total carbohydrate content, namely the glycemic load and the relative composition of the meal, evaluating the proportion of the other three macronutrients (fat, protein, fiber), as well as the way of cooking, the use of spices and meal timing [425,426,427,428].

The personalization of the nutritional support for the infertile patient is crucial when it comes to glucose homeostasis. It is generally suggested to “eat at least five portions of fruit and vegetables daily”. However, these general guidelines do not suit all patients, particularly those with abnormal glucose homeostasis, who are highly represented in the infertile population. Indeed, fructose, when consumed alone and far from the main meals (as often suggested), leads to a rapid increase in glycemia since it is readily absorbed and rapidly metabolized by the liver [429]. On the one hand, this results in increased food-seeking and sugar cravings, while on the other hand, it increases both fat production and storage [430]. Although fructose does not acutely increase insulin levels, a chronic exposure indirectly causes hyperinsulinemia and obesity [429]. Chronic exposure seems associated with hepatic inflammation and cellular stress (oxidative and endoplasmic) [431] and with increased cortisol production [432], all affecting fertility.

Another general suggestion about glucose homeostasis is to “avoid pasta at dinner”, especially when weight loss represents a goal of the nutritional program. However, some studies support the notion that consuming carbohydrates at dinner is beneficial for individuals suffering from insulin resistance. In these subjects,“pasta at dinner” promotes greater weight loss and more favorable changes in leptin, ghrelin, and adiponectin concentrations, as well as greater improvements in *C-*reactive protein, tumor necrosis factor-alpha, and IL-6 levels [433,434]. This is a further indication that nutritional suggestions/guidelines should always be tailored to the specific case and always consider the endocrine status of the patient.

### 5.4. Consumption of Foods with Antioxidant Properties

A large body of evidence suggests that an antioxidant status can be modulated by diet. Luckily enough, several foods exert both antioxidant and anti-inflammatory activities [435]. A balanced consumption of whole grains, supporting a low glycemic index, and meals [436] enriched with polyphenols (e.g., quercetin, naringenin, epigallocatechin gallate, resveratrol) [437], flavonoids (e.g., silymarin) [438] and carotenoids (found in plant foods, herbs, and spices), represents the best diet [435,439,440]. Furthermore, it is possible to support the activity of antioxidant enzymes with an appropriate intake of selenium, zinc, and vitamins (particularly E and C), which act as cofactor for those enzymes [441]. Furthermore, both anti-inflammatory and antioxidant properties have been associated with the intake of omega-3 fatty acids [442,443], EVOO [444], garlic [445], fermented milk, and oilseeds [436,446]. Some other important bioactive compounds exerting a strong protective antioxidant effect are curcumin [447,448] and capsaicin [449]. Finally, the supplementation with probiotics and vitamin D [442], α-lipoic acid [450], and melatonin could be beneficial for women’s fertility, as also demonstrated by a study investigating pregnancy outcomes [451].

## 6. Conclusions and Future Perspectives

Growing evidence unveils how both healthy lifestyle habits and dietary patterns favor reproductive success. However, a specific “fertility diet” has not been yet identified and probably never will be. This is because each person is a unique individual, with its own genome, proteome, metabolome, microbiome, and exposome. This should always be addressed when choosing a nutritional approach, as for any therapy in medicine. In line with the concept of personalized medicine, nutritional support for the infertile patient should be tailored to the individual, aiming at “precision nutrition”. Since several diseases linked to infertility are related to a proinflammatory state, with the dysregulation of important markers of inflammation (e.g., CRP, TNF-α, NF-kB, and IL-6), this aspect should always be considered in the nutritional management of infertile patients, elaborating diets with a marked anti-inflammatory signature [80]. These diets, for a more complete anamnesis, should be based on the assessments of various parameters such as self-reported dietary and lifestyle habits and anthropometric data and should be combined with the genetic profile and gut status as assessed via microbiomics and metabolomic approaches (Figure 2). Only a comprehensive view of all these aspects and their integration in the full picture could allow a more effective nutritional intervention in the management of female infertility. Although the genotype may predispose to nutrition-related disease, dietary factors may influence gene expression via their epigenetic activity, thereby affecting proteins and metabolites [26]. In our view, the nutritional management of infertile patients should be tailored to each patient’s characteristics, keeping in mind the strong relationship between infertility and common chronic noncommunicable diseases (NCDs). Infertile women are, in fact, more subject to premature mortality due to cancer (i.e., digestive organs, genito-urinary organs, and lymphatic and hematopoietic tissue) and non-malignant diseases of the gastrointestinal system [452]. Therefore, nutritional support in infertile patients is even more important due to its potential long-term protective effect [453,454].

This paper summarized some of the strategies to tailor personalized nutritional support in infertile women, mainly focusing on the management of chronic low-grade inflammation, a condition that characterizes different reproductive disorders. Of note, infertility affects men as well, and male factor infertility accounts for 50% of causes [455]. Therefore, a limit of this paper is that it considers only the nutritional management of the female counterpart, yet mounting evidence demonstrates that personalized nutritional support would be useful also for the male counterpart [456,457,458].

Considering the evidence at hand, the personalized nutritional support for female fertility should always:Consider the -omic characteristics of each patient (e.g., genotype, microbiome);Deepen the patient’s life choices (e.g., vegetarianism) to outline a more appropriate supplementation;Combine several nutrients with anti-inflammatory nutrigenomics properties as they may establish synergies and/or modulate several cellular and molecular pathways at once;Exclude proinflammatory foods or habits (e.g., harmful cooking methods) because often it is not only a question of “what to eat” but also “what to avoid”;Monitor the postprandial glycemic and insulinemic response, which figure among the main mechanisms by which diet can affect fertility;Carefully manage the glycemic load of each meal, the combination of the foods together with the way of cooking, the use of spices, and meal timing.

Personalized nutrition is a tool to preserve health rather than treat a condition. Since infertility is a social problem and an emerging priority for public health [459], we think that a change in the cultural mindset is required and that healthier and personalized nutrition shall be suggested earlier and be continued throughout life in order to prevent infertility, rather than to treat it [253].

## Figures and Tables

**Figure 1 nutrients-14-01918-f001:**
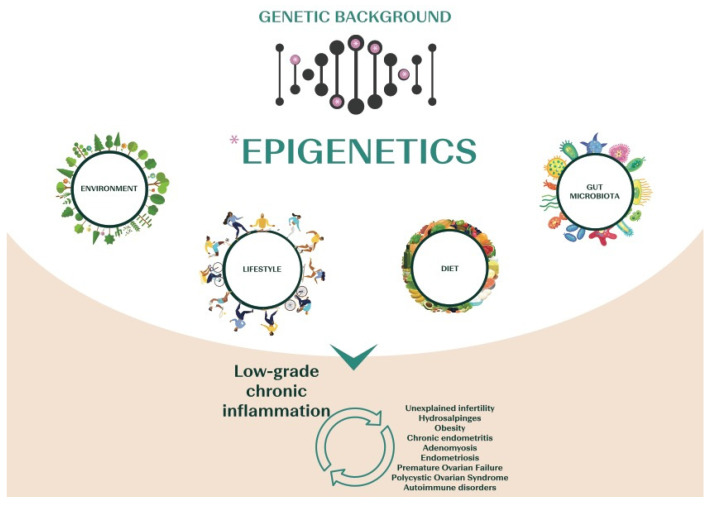
Scheme of the main causative factors of chronic low-grade inflammation, a state that characterizes several infertility-related diseases.

**Figure 2 nutrients-14-01918-f002:**
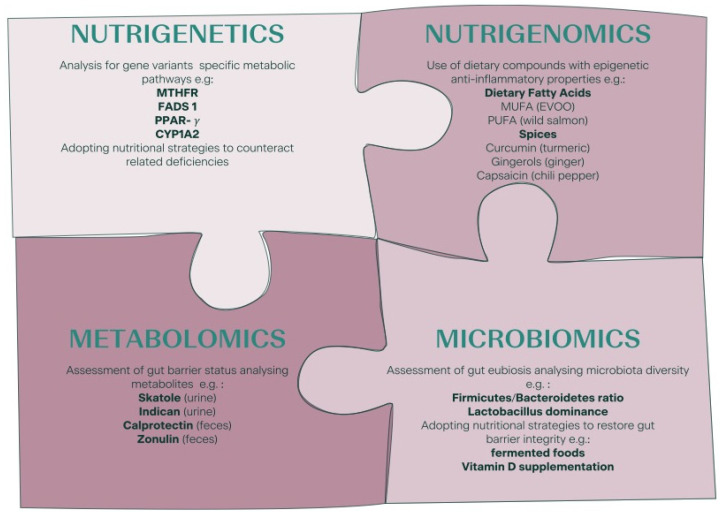
Personalized nutrition in the management of female infertility: practical examples of nutrigenetics, nutrigenomics, microbiomics, and metabolomics aspects to consider. Methylenetetrahydrofolate Reductase (*MTHFR*); Fatty Acid Desaturase 1 (*FADS 1*); peroxisome proliferator-activated receptor gamma (*PPAR*-*γ*); Cytochrome P450 1A2 (*CYP1A2*); Monounsaturated Fatty Acids (MUFA); Polyunsaturated Fatty Acids (PUFA).

**Table 1 nutrients-14-01918-t001:** Practical examples of single nucleotide polymorphisms (SNPs) that can influence a proinflammatory environment in infertile women and the suggested nutritional intervention.

Genes/ Haplotypes	Nutrition and Health Pattern Involved	SNPs	Genotype Differences				Nutritional Intervention in Subjects at Risk
**MTHFR**	**Folate metabolism**	rs1801133	C/C normal enzyme activity	C/T reduced enzyme activity	T/T reduced enzyme activity		Adequate B vitamin-enriched diets (green raw vegetables, fruits, shellfish, etc.) and/or adequate supplementation (wildtype 200 μg/day; intermediate 400 μg/day; risk 800 μg/day; [101]) with adequate B6, B12 and choline intake
		rs1801131	A/A normal enzyme activity	A/C reduced enzyme activity	C/C reduced enzyme activity		
**PEMT**	**Choline metabolism**	rs7946	G/G normal enzyme activity	A/G higher choline deficiency risk	A/A higher choline deficiency risk		Increased amount of folate rich foods (raw green leafy vegetables, seeds, fruits) [103]
		rs12325817	G/G normal choline metabolism	C/G higher choline deficiency risk	G/G higher choline deficiency risk		
**MTHFD1**		rs2236225	G/G normal choline metabolism	A/A higher choline deficiency risk	A/G higher choline deficiency risk		
**FTO**	**Obesity, fat mass and Met-S associated genes**	rs9939609	T/T lower risk of obesity and adiposity	A/T higher risk of obesity and adiposity	A/A higher risk obesity and adiposity		Hypocaloric MedDiet in general with low saturated fats and limited carbohydrates [108,109,110,111,112]. Higher intake of proteins is recommended in risk allele carriers [113]
		rs1558902	T/T lower risk of obesity and adiposity	A/T intermediate risk of obesity and adiposity	A/A higher risk obesity and adiposity		
**LEP**		rs2167270	G/G lower risk of obesity and IR	G/A higher risk of obesity and IR	A/A higher risk of obesity and IR		Hypo/normo-caloric diet with reduced SFA and carbohydrates intakes especially from sweets and snacks [114]
		rs7799039	G/G lower risk of obesity and IR	G/A higher risk of obesity and IR	A/A higher risk of obesity and IR		
**ADIPOQ**		rs266729	C/C normal adiponectin levels, lower risk of Met-S	C/G diminished adiponectin levels, higher Met-S traits	G/G diminished adiponectin levels, higher Met-S traits		Reduced SFA intake [115]
**LCT**	**Lactose metabolism**	rs4988235	T/T lactase persistence	C/T intermediate phenotype	C/C lactose intolerance		Diet low in lactose (<12 g) use of fermented dairy products and/or adequate lactase [116] and probiotic supplementation [117]
**FADS1**	**Long-fatty acids synthesis**	rs174537	C/C normal biosynthesis	C/T reduced biosynthesis	T/T impaired biosynthesis		Adequate apport of foods containing omega-3 PUFAs and/or adequate omega-3 supplementation [118]
		rs174547	T/T normal D5D and D6D fatty acid desaturase enzyme activity	T/C decreased D5D and D6D fatty acid desaturase enzyme activity	C/C decreased D5D and D6D fatty acid desaturase enzyme activity		
**PPAR-G**	**Glucose metabolism/diabetes or insulin resistance risk**	rs1801282	G/G reduced risk of T2DM and IR	G/C intermediate risk of T2DM and IR	C/C Increased risk of T2DM and IR		According to the combination of genetic risk: low glycemic index diet with the characteristics of the MedDiet so adequate fiber intake (30 g/day), limitation of refined carbohydrates and replacement of animal fats with vegetable ones, especially MUFAs (extra virgin olive oil) but also PUFAs (oily fruit). Possibly support with omega3 supplementation. [119,120,121,122,123,124,125]
**TCF7L2**		rs12255372	G/G lower risk of T2DM and gestational diabetes	G/T higher risk of T2DM and gestational diabetes	T/T higher risk of T2DM and gestational diabetes		
		rs7903146	C/C normal insulin response	C/T intermediate insulin response	T/T impaired insulin response		
**KCNJ11**		rs5219	E/E normal glucose tolerance, lower risk of T2DM and IR	E/K intermediate risk of T2DM and IR	K/K altered glucose tolerance, higher risk of T2DM and IR		
**CYP1A2**	**Caffeine metabolism**	rs762551	A/A fast metabolizer	A/C slow metabolizer	C/C slow metabolizer		Caffeine intake <100 mg/day [126]
**HLA**	**Celiac disease predisposition and gluten sensitivity**	rs2395182 rs7775228 rs2187668 rs4639334 rs7454108 rs4713586	**DQ2/DQ8-negative**	**Half DQ2-positive** HLA-DQA1*0501 or 0505 or HLA-DQB1*0201 or 0202	**DQ2-positive** HLA-DQA1*0501 or*0505 and HLA-DQB1*0201 or *0202	**DQ8-positive** HLA-DQA1*03 and HLA-DQB1*0302	Gluten-reduced diet (from 3 g up to 13 g) [127] or gluten-free diet using naturally GF products (e.g., rice, quinoa, amaranth, buckwheat)

Methylenetetrahydrofolate Reductase (MTHFR); Phosphatidylethanolamine *N-*Methyltransferase (PEMT); methylenetetrahydrofolate dehydrogenase 1 (MTHFD1); Fat Mass and Obesity-Associated (FTO); Leptin (LEP); Adiponectin (ADIPOQ); Lactase (LCT); Fatty Acid Desaturase 1 (FADS 1); Peroxisome Proliferator-Activated Receptor Gamma (PPAR-*γ*); Transcription Factor 7 Like 2 (TCF7L2); Potassium Inwardly Rectifying Channel Subfamily J Member 11 (KCNJ11); Cytochrome P450 1A2 (CYP1A2); Human Leukocyte Antigen (HLA); Mediterranean Diet (MedDiet); Insulin Resistance (IR); Metabolic Syndrome (Met-S); Type 2 Diabetes Mellitus (T2DM); Saturated Fatty Acids (SFA); Monounsaturated Fatty Acids (MUFA); Polyunsaturated Fatty Acids (PUFA);Gluten-Free (GF).

## Data Availability

Not applicable.

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
