# Peer review of "Personalized Nutrition in the Management of Female Infertility: New Insights on Chronic Low-Grade Inflammation"

_nutrients, 2022, doi:10.3390/nu14091918_

Round 1

Reviewer 1 Report

General Comments: This is a comprehensive review attempting to summarise the literature related to nutrition and female infertility. For the most part this has been done well, however there are some sections of the review where the link between nutritional aspects and female infertility were not made well or at all, therefore these sections relevance to the topic is questionable (see specific comments).  Finally, there are quite few typographical and English expression errors which need to be fixed.

Specific Comments

P2, line 54 "This is because either poor or excessive intake of proteins, vitamins, micro and macro minerals alter energy balance, which is directly correlated to reproductive performance" Should you also consider adding CHO and lipid intake to this list?

P2 line 76. "...patients seeking for a pregnancy." should read "seeking a pregnancy"

P3, line 108. "...effective than a general dietary advice." Delete the "a".

P3, line 111. "Eventually, several hormonal unbalances are involved". Should eventually be changed to evidently? Change unbalances to inbalances.

P3, line 141. "...patients affect from polycystic" should read "patients affected with polycystic"

P3, line 146. "affected from endometriosis" change to "affected with endometriosis"

P4, line 155. "..have been reported increased " change to  " are increased"

P4 line 155-6. "...affected from adenomyosis," change to "affected with adenomyosis,"

P4 line 163. "..factors of chronic low grade inflammatory," change to "factors of chronic low grade inflammation,"

P4 paragraph beginning line 172. Suggest that this paragraph be deleted as no connection is made back to how this relates to female infertility. Alternatively, make the connection.

P4, line 179. "..may impair also folliculo-" delete "also".

p5 line 224. "..Genetic tests shall be preferentially" change shall to should.

P6, line 273. "...A1298C, at last, is" delete- , at last,

P 6, lines 291-292. "...(wildtype 200μg/die, intermediate 400μg/die 291 and risk 800μg/die) - change die to day. Same error is also in Table 1 and P7 line 294, P12, line 579, P13, line 601.

P11, line 496. "...Not only by the omega3-omega6 ratio" delet "by".

P11, line 503. "...can help identifying patients" change to "can help identify patients"

P13, line 585. "...refer to paragraph 5.3" change to " refer to section 5.3"

P13, line 599. "...caffein," change to "caffeine,"

P14, line 659. "...reduces the risk obesity" change to "reduces the risk of obesity"

P14, line 667. "...mandatory prior conception" change to "mandatory prior to conception"

Sections 3.2 and 3.3. How do these sections link to female infertility. Either clarify the link or delete from review.

P17, line 802. "...and of corticosteroids" delete "of".

P17, line 821. "...in cytokines production" change to "in cytokine production"

P17, line 825. "...As a matter of fact, in fact," change to " In fact, ..."

P17, line 831. "..Based on all these evidence," change to " Based on all this evidence,"

P17, line 838. "...are of particularly important." change to "are of particular importance."

P20, line 988-990. reword this sentence, it does not make sense.

P20, line 994. "...in paragraph 2.1.6" change to "in section 2.1.6".

P21, line 1039. "...investigating pregnancies outcomes" change to "investigating pregnancy outcomes"

Author Response

Authors: We want to sincerely thank the reviewers and Editors for their time and constructive comments. We took this opportunity to improve the quality of our manuscript by further underlining both its limitations and strengths. Please see in attachment our point-by-point response.

Reviewer 2 Report

Thank you for giving me the opportunity for review the manuscript entitled “Personalized nutrition in the management of female infertility: new insights on chronic low-grade inflammation.”

The infertility emerges as an epidemic problem of the modern civilization.  Given the scale of the problem, the material presented is important and relevant.

The manuscript is interesting and in scope of the Journal however it requires some clarifications.

To enrich the work, it should be more strongly emphasized that infertility affects both women and men. Additionally, women's consumption of e.g. alcohol and supplements should be considered.

Author Response

(The authors gave the same response as above.)
